# SARS-CoV-2 variants with reduced infectivity and varied sensitivity to the BNT162b2 vaccine are developed during the course of infection

**Dina Khateeb[1], Tslil Gabrieli[1], Bar Sofer[1], Adi Hattar[1], Sapir Cordela[1], Abigael Chaouat[2], Ilia Spivak[3], Izabella Lejbkowicz[4], Ronit Almog[4], Michal Mandelboim[5], Yotam Bar-On[1]***

**1** Department of Immunology, Rappaport Faculty of Medicine, Technion-Israel Institute of Technology, Haifa, Israel, **2** The Concern Foundation Laboratories at the Lautenberg Center for Immunology and Cancer Research, Institute for Medical Research Israel Canada (IMRIC), The Hebrew University Hadassah Medical School, Jerusalem, Israel, **3** Department of Pediatrics B, Ruth Rappaport Children's Hospital, Rambam Health Care Campus, Haifa, Israel, **4** Epidemiology Unit and Biobank, Rambam Health Care Campus, Haifa, Israel, **5** Central Virology Laboratory, Sheba Medical Center, Tel Hashomer, Israel

\* ybaron@technion.ac.il

**Data Availability Statement:** All relevant data are within the manuscript and its Supporting Information files.

## Abstract

In-depth analysis of SARS-CoV-2 quasispecies is pivotal for a thorough understating of its evolution during infection. The recent deployment of COVID-19 vaccines, which elicit protective anti-spike neutralizing antibodies, has stressed the importance of uncovering and characterizing SARS-CoV-2 variants with mutated spike proteins. Sequencing databases have allowed to follow the spread of SARS-CoV-2 variants that are circulating in the human population, and several experimental platforms were developed to study these variants. However, less is known about the SARS-CoV-2 variants that are developed in the respiratory system of the infected individual. To gain further insight on SARS-CoV-2 mutagenesis during natural infection, we preformed single-genome sequencing of SARS-CoV-2 isolated from nose-throat swabs of infected individuals. Interestingly, intra-host SARS-CoV-2 variants with mutated S genes or N genes were detected in all individuals who were analyzed. These intra-host variants were present in low frequencies in the swab samples and were rarely documented in current sequencing databases. Further examination of representative spike variants identified by our analysis showed that these variants have impaired infectivity capacity and that the mutated variants showed varied sensitivity to neutralization by convalescent plasma and to plasma from vaccinated individuals. Notably, analysis of the plasma neutralization activity against these variants showed that the L1197I mutation at the S2 subunit of the spike can affect the plasma neutralization activity. Together, these results suggest that SARS-CoV-2 intra-host variants should be further analyzed for a more thorough characterization of potential circulating variants.

**Funding:** Y.B received the ISF-2029450 grant from the Israel Science Foundation https://www.isf.org.il/#/ The funders had no role in study design, data collection and analysis, decision to publish, or preparation of the manuscript.

**Competing interests:** The authors have declared that no competing interests exist.

## Author summary

The global employment of mRNA-based COVID-19 vaccines, together with their high protection, has enabled to control the pandemic in places with high vaccination rate. However, the recent increase in the frequency of SARS-CoV-2 variants with reduced sensitivity to antibody neutralization has raised concerns about the ability of COVID-19 vaccines to eradicate SARS-CoV-2. Despite in-depth characterization of such variants, little is known about SARS-CoV-2 evolution within the host. Here, we preformed in-depth analysis of swab samples isolated from infected individuals and demonstrated that intra-host variants with reduced infectivity are developed during the course of the infection. Furthermore, these variants have varied sensitivity to the BNT162b2 vaccine and to convalescent plasma. In addition, we identified a point mutation at the S2 subunit of the spike protein, which impairs antibody neutralization activity. Our findings shed light on SARS-CoV-2 adaptation to the human host and highlight the importance of mapping the landscape of SARS-CoV-2 intra-host variability for better evaluating variants of concern.

## Introduction

SARS-CoV-2 has a relatively large genome in comparison with other RNA viruses such as HIV-1 and influenza virus [1,2]. In order to enhance the replication fidelity of its 32 kilobases genome, SARS-CoV-2 has evolved to encode a proof-reading machinery with 3'-5' exonuclease activity [3]. Nevertheless, since the initial SARS-CoV-2 outbreak in Wuhan, the virus has acquired several mutations that affected its infectivity and immunogenicity [4–6]. Global swab sampling and sequencing of bulk RNA from infected individuals has allowed to evaluate the inter-host SARS-CoV-2 variability while sequencing databases such as Global Initiative for Sharing All Influenza Data (GISAID) and Nextstrain allow real-time evaluation of SARS-CoV-2 inter-host variability and facilitate tracking of SARS-CoV-2 global spread [7]. A few circulating SARS-CoV-2 variants have been the focus of extensive research due to their rapid spread and high infectivity [8]. These include the Alpha variant (B.1.1.7/501Y.V1), the Beta variant (B.1.351/501Y.V2), the Gamma variant (P.1) and the Delta variant (B.1.617.2) [9]. As mRNA-based SARS-CoV-2 vaccines are deployed globally and show high efficacy in preventing infection, the question of whether a vaccine resistance variant will emerge is ongoing. Notably, several reports demonstrated that neutralizing antibodies elicited by vaccines show reduced activity against viruses that carry specific spike mutations that are present in the Alpha, Beta, Gamma and Delta variants and that these variants are less effectively neutralized by plasma from vaccinated individuals [9,10]. This is of special interest, since the emergence of such variants with reduced sensitivity to antibodies was shown to be detrimental to the attempts to develop a universal vaccine that will protect against highly mutable RNA viruses [11,12].

To dissect the effect of vaccination, antibody treatment and the host's selective pressure on the landscape of viral quasispecies, single-genome sequencing (SGS) techniques were developed [13,14]. This strategy enabled in-depth analysis of viral mutation that occurs during infection and has been widely used to study HIV-1 host adaptation [13–16]. SGS was shown to accurately recapitulate the *in vivo* frequency of HIV-1 mutant strains without *in vitro* culturing bias and enable the detection of rare, mutated viruses [15–19]. Thus, this method facilitated the identification and tracing of antibody-resistance HIV-1 varinats that emerged during natural HIV-1 infection in response to immune pressure [15,17,19]. Based on such genomic analysis, the mechanism of HIV-1 escape from various broadly neutralizing antibodies was identified and was later used to demonstrate that a dual antibody treatment where a combination of two broadly neutralizing

antibodies are infused simultaneously can limit HIV-1 escape and lead to sustained viral suppression [16,18]. Interestingly, despite a much lower mutation rate, recent studies showed that SARS-CoV-2 has the potential to escape neutralizing antibodies, namely by introducing the E484K, N501Y or K417N/E484K spike mutations [8–10,20,21].

As mRNA-based SARS-CoV-2 vaccines became widely available, circulating SARS-CoV-2 variants are being closely monitored for amino acid changes in the spike, for their infectivity and for their sensitivity to the vaccines [8,10,20]. These evaluations have made it possible to better predict possible surges in COVID-19 cases and facilitated a better control of the pandemic [22]. However, these analyses have several limitations. First, most of them focus on variants with high infection rates such as the Gamma and Delta variants, while several previous studies indicated that viral escape from neutralizing antibodies often leads to reduced infectivity of the mutated virus [23–26]. Second, in-depth characterization of specific viral mutation was done mainly on mutations that were found in variants that are frequently circulating between individuals [8–10,20]. Thus, less is known about the virological and immunological characteristics of intra-host SARS-CoV-2 variants that are developed in the respiratory system during the course of infection.

In this study, we preformed SGS for SARS-CoV-2 spike (S) gene and nucleocapsid (N) gene derived from swab samples isolated from infected individuals. Using this approach, we were able to isolate and characterize SARS-CoV-2 variants from all individuals who were analyzed. Most of the SARS-CoV-2 variants that we have isolated were not identified by previous bulk sequencing approaches, are rarely documented in SARS-CoV-2 genome databases, and showed reduced infectivity when incubated with ACE2-expressing cells. Notably, these intra-host variants also showed different degrees of sensitivity to the BNT162b2 vaccine and to convalescent plasma.

## Results

### Single-genome sequencing of swab samples from infected individuals

To isolate and analyze intra-host SARS-CoV-2 variants, nine individuals who were positively tested for SARS-CoV-2 were chosen from a cohort of infected individuals who were diagnosed at the Central Virology Laboratory Sheba Medical Center. Individuals were chosen based on age and on the estimated geographic infection region in order to minimize oversampling of specific age group or geographic regions (Fig 1A). Genomic material was isolated from nose-throat swabs after inactivation, and the SARS-CoV-2 copy number was calculated by correlating qRT–PCR and cycle threshold (Ct) values with standard samples of known viral RNA copy number [27]. Swab samples of all nine individuals showed relatively high viral load ranging from ~3.4x$10^5$ copies/ml to ~5.5x$10^5$ copies/ml (Fig 1B). Next, we performed SGS of SARS-CoV-2 isolated from infected individuals. To this end, RNA isolated from swab samples was subjected to cDNA synthesis using SARS-CoV-2-specific primer (Fig 1C and S1 Table). The cDNA was then diluted to a single copy per well, and the SARS-CoV-2 S gene and N gene were amplified by a nested PCR reaction using two sets of gene-specific primers (Fig 1C and S1 Table). All PCR products were analyzed in gel electrophoresis (S1 Fig) and were sequenced by llumina MiSeq Next Generation Sequencer. Using this approach, we isolated and sequenced an average of 44 SARS-CoV-2 genomes from each of the nine individuals.

### Spike and nucleocapsid variants are developed during the course of SARS-CoV-2 infection

Specific analysis of S-gene sequences isolated from individual 3804 showed no intra-host variability, as all 46 swab-derived viruses expressed an identical S gene. Interestingly, SARS-CoV-2

**A**

| Patient ID | Gender | Age | Estimated infection place |
|---|---|---|---|
| 3120 | Female | 74 | Spain/Switzerland |
| 3804 | Female | 35 | Spain |
| 3807 | Female | 34 | Austria |
| 3953 | Male | 43 | Germany/Greece |
| 4000 | Female | 11 | Spain |
| 3380 | Male | 40 | Germany/Greece |
| 4065 | Male | 31 | USA/Russia |
| 48552 | Female | 73 | Israel |
| 48559 | Male | 51 | Israel |

**B**

**C**

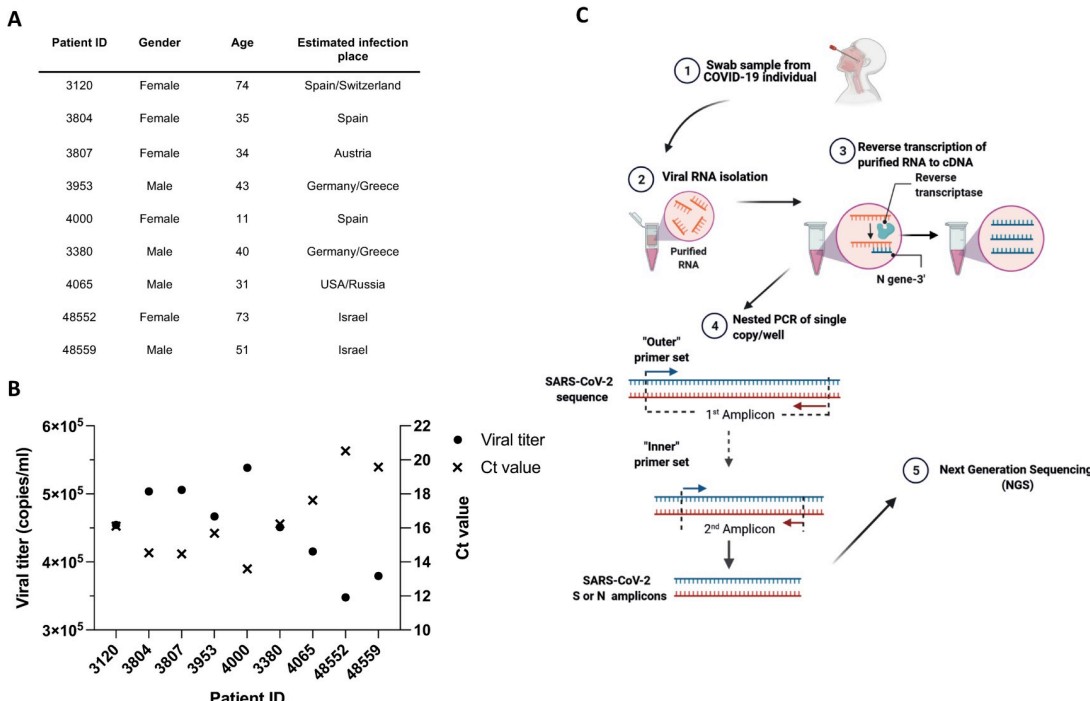

**Fig 1. Study design and participants demographics.** (A) Detailed description of SARS-CoV-2-infected individuals from whom nose-throat swabs were isolated. The infected individuals are represented by the study ID. The geographic location where the individuals were infected is shown in the table. (B) Ct values (right y-axis) and viral titers (left y-axis) of SARS-CoV-2 in nose-throat swabs. Values were obtained by Real-Time Reverse Transcriptase-Polymerase Chain Reaction measurements that was performed during the clinical diagnostic of the swabs. The study ID of the infected individuals is shown in the x-axis. (C) Schematic representation SARS-CoV-2 SGS. RNA isolated from nose swabs was reverse transcribed into cDNA using SARS-CoV-2 specific primer (N-Out-3'). cDNA was diluted to single copy per well and subjected to a nested PCR to amplify single copies of SARS-CoV-2 S gene and N gene. PCR products were then sequenced by Illumina MiSeq. Figure was generated using biorender.com.

intra-host variants were found in all other individuals analyzed. The frequency of SARS-CoV-2 intra-host variants ranged between 6% (individual 4000) and 24% of the isolated viruses (individual 3807, Fig 2). Although the abundancy of each intra-host mutant was sparse, we also found a few S-gene mutations that were repeated in several viral isolates from individual 3807 (nucleotide substitution A3300G and T1389C, Fig 2) and one mutation (C968T) repeated in two individuals (Individual 4000 and 48559). Further analysis of SARS-CoV-2 sequences from individual 3120 revealed S-gene variants with nonsense mutations (C1687T) in which a glutamine amino acid was replaced with a premature stop codon. An additional premature stop codon was also observed in individuals 3380 and 3807 (Figs 2 and S2A). Both of these S-gene variants encode for a spike protein with an intact receptor binding domain (RBD) but are missing either the entire S2 subunit (S-gene isolated from individuals 3120 and 3380) or most of it (S-gene isolated from individual 3807). Interestingly, the A3029G mutation that appears in individual 48552 leads to amino acid substitution Q110R that was previously associated with SARS-CoV-2 escape from polyclonal antibodies [28]. In addition, a variant that was identified in two separate individuals (individual 4000 and individual 48559) carried a mutation in the N-glycosylation site (C968T, T323I). An additional mutation in an O-glycosylation site was identified in individual 3953 (A1807G, N603D).

Finally, to further validate our results and to verify that the mutations we identified are not a result of PCR-mediated mutations, we single-genome amplified 55 genomes of WT Whan-

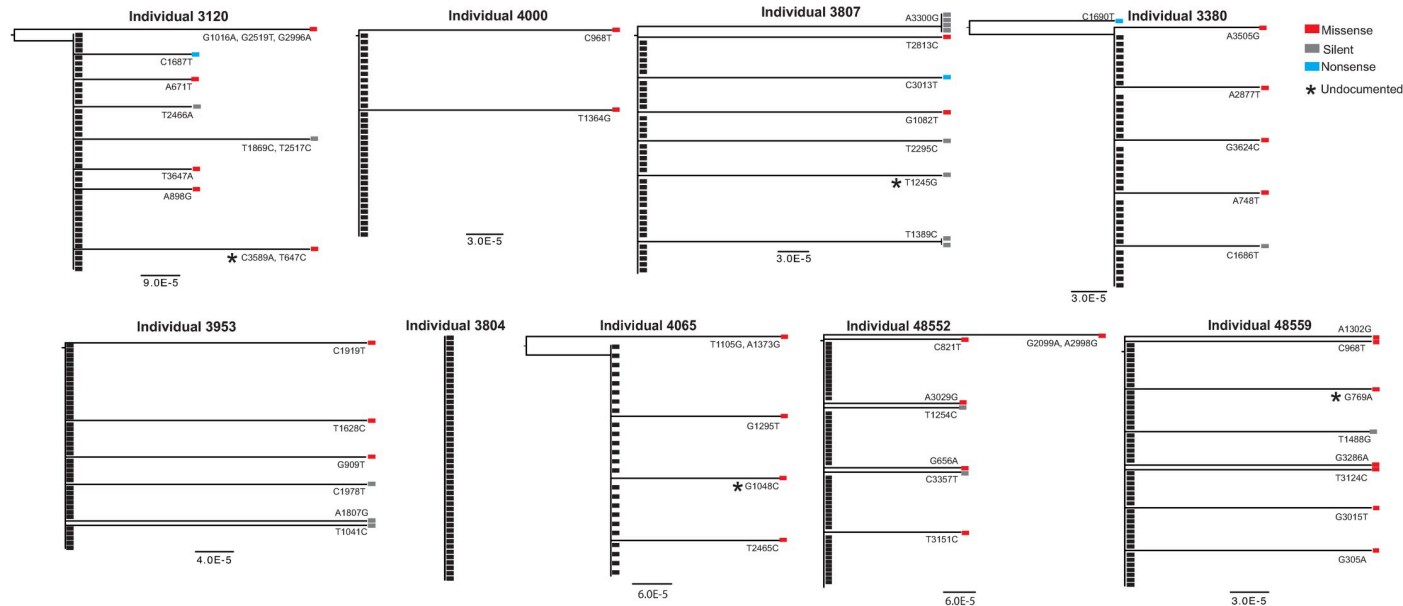

**Fig 2. Single genome analysis reveals SARS-CoV-2 spike diversity.** Phylogenetic trees depicting S genes isolated from nose-throat swabs by SGS. IDs of individuals from whom swabs were isolated are indicated in the figure. Black rectangles depict the dominant S gene isolated from each swab sample. Red rectangles depict S genes with missense mutations, grey rectangles depict S genes with silent mutations and blue rectangles depict S genes with nonsense mutations. Mutations that are not documented in the GSAID are marked with an asterisk. For each S gene variant, the nucleotide substitution is indicated (based on Wuhan-Hu-1 numbering). The segment with the number below each phylogenetic tree shows the length of branch that represents an amount genetic change. The amount of genetic change is the number of nucleotide substitutions divided by the length of the spike sequence.

Hu-1 spike, which equals the maximum number of genomes isolated from an infected individual and observed no mutations in all of the amplified genomes (S3 Fig).

We also compared the SGS data derived from individuals 3120, 4000, 4065, 3380 and 3807 to sequencing results that were obtained when bulk PCR reactions were performed. Variant calling analysis was used to identify single nucleotide polymorphism (SNP) for each sample that was amplified by bulk PCR. In accordance with the SGS results, the variant calling analysis identified that all 5 individuals were infected with the A1841G (D614G variant) and also identified the A3300G (T1100) in individual 3807 (S2 Table). However, all the other intra-host variants from these 5 individuals were not identified by the variant calling analysis, indicating that a higher sequencing coverage is required to identify these variants, as in bulk PCR reaction, exponential amplification of the high-frequency variant might dilute the low frequency clones. Of note, this comparison also revealed that some variants were missed by our SGS analysis, as several new variants appeared in the variant calling analysis (S2 Table).

Next, the frequency of the S-gene mutations isolated by our SGS strategy was assessed from GISAID database, which documents the global SARS-CoV-2 variants isolated by bulk sequencing [29]. Most of the mutations we retrieved by SGS were absent from the current sequencing databases or found in very low frequencies (S4A Fig). Thus, the SGS of SARS-CoV-2 S genes facilitated the identification of rare intra-host SARS-CoV-2 variants that are developed during the course of SARS-CoV-2 infection.

To further characterize SARS-CoV-2 intra-host variability, SARS-CoV-2 N-gene sequences were obtained by SGS of swab-derived viruses. We chose to analyze N-gene sequences since it was previously shown that the N protein is the most abundant viral protein in infected cells and is highly immunogenic [30–32]. Thus, we postulated that the N-gene will be subjected to extensive immune selective pressure. A total of 142 N-gene sequences were isolated from five

infected individuals out the nine individuals recruited for this study, from whom sufficient amounts of viral RNA were isolated to preform analysis to both the S-gene and the N-gene. Despite the relative short size of the SARS-CoV-2 N gene (1260 bp) in comparison with the 3822 bp S gene, we detected intra-host N-gene variants in all five individuals analyzed (Figs 3A and S2B). Interestingly, an identical N-gene variant C168G was isolated from four different individuals (individual 3953, individual 3120, individual 3807 and individual 4000; Fig 3A and 3B). This variant was also not found in sequences deposited in the GISAID database (S4B Fig). Similar to our S-gene analysis, most of the N-gene variants we have identified were yet to be documented, but the synonymous nucleotide substitution (G609A) we have identified was found in numerous SARS-CoV-2 isolates (S4B Fig). The abundancy of N-gene mutants varied between the tested individuals, and the highest degree of N-gene variability was observed in individual 3120, who is the oldest among all individuals analyzed (74 years old). However, the degree of N-gene variability did not correlate with the degree of S-gene variability. Thus, correlation of intra-host SARS-CoV-2 variability with age, viral load and disease symptoms should be further addressed using a larger cohort of infected individuals.

A recent report demonstrated that N mutations found in circulating variants can improve viral particle assembly and luciferase expression in pseudovirus-infected cells [33]. Thus, we chose all N-variants with amino acid substitution (missense mutations) identified by our SGS analysis to generate pseudoviruses that carry WT or mutated N-protein. N-gene expression was verified by PCR in the transfected cells (S5 Fig) and pseudoviruses were collected from the media. By measuring the p24 levels in the cell supernatant and the luminescence in the infected cells, we noted that the N-protein mutations A119V, M411I, A414P and E62D increase the pseudovirus production and the luciferase expression in the infected cells (Fig 3C and 3D).

## Spike intra-host variants show reduced infectivity capacity

Our single-genome analysis revealed that the intra-host variants we have identified are present in low frequencies in the respiratory system and that a more dominant and abundant SARS-CoV-2 variants could be identified in these swab samples. Thus, we postulated that these variants would show reduced infectivity and replication capacity. To test this, we analyzed the effect of the spike mutations we have identified by our SGS analysis on SARS-CoV-2 capacity to infected ACE2-expressing cells. We selected ten spike variants with mutations that are located at different domains of the spike protein for further analysis: L303F, K300E and L216P mutations at the S1 N-Terminal Domain (NTD), T323I and C361F mutations located at the receptor binding domain (RBD(, L938P mutation located at heptad repeat 1 (HR1) and L1197I mutation at heptad repeat 2 (HR2) at S2, two other variants from undefined domains at the S1 and S2 subunits, F543S and C840F, respectively. The tenth variant carries two mutations, G339D at the RBD and G909T at the S2 subunit (Fig 4A). These variants were selected based on the mutations location in the spike protein to verify that variants with mutations at different spike domains are further analyzed. Then we have generated SARS-CoV-2 pseudoviruses that carry these mutated spike proteins and a NanoLuc luciferase reporter gene (Fig 4A) and verified their production by measuring p24 levels (Fig 4B). To test the infectivity capacity of the SARS-CoV-2 pseudoviruses that express the mutated spike proteins, we first generated a stable 293T-ACE2 cell line and verified the membrane expression of ACE2 with polyclonal anti-ACE2 antibody and RBD-Ig (Fig 4C). Significant increase in luminescence values were seen when SARS-CoV-2 pseudoviruses were used to infect 293T-ACE2 cells while no significant luminescence values were seen when bald particles with no spike proteins were used for infection (Figs 4D and S6). Importantly, all ten spike mutants that we have tested

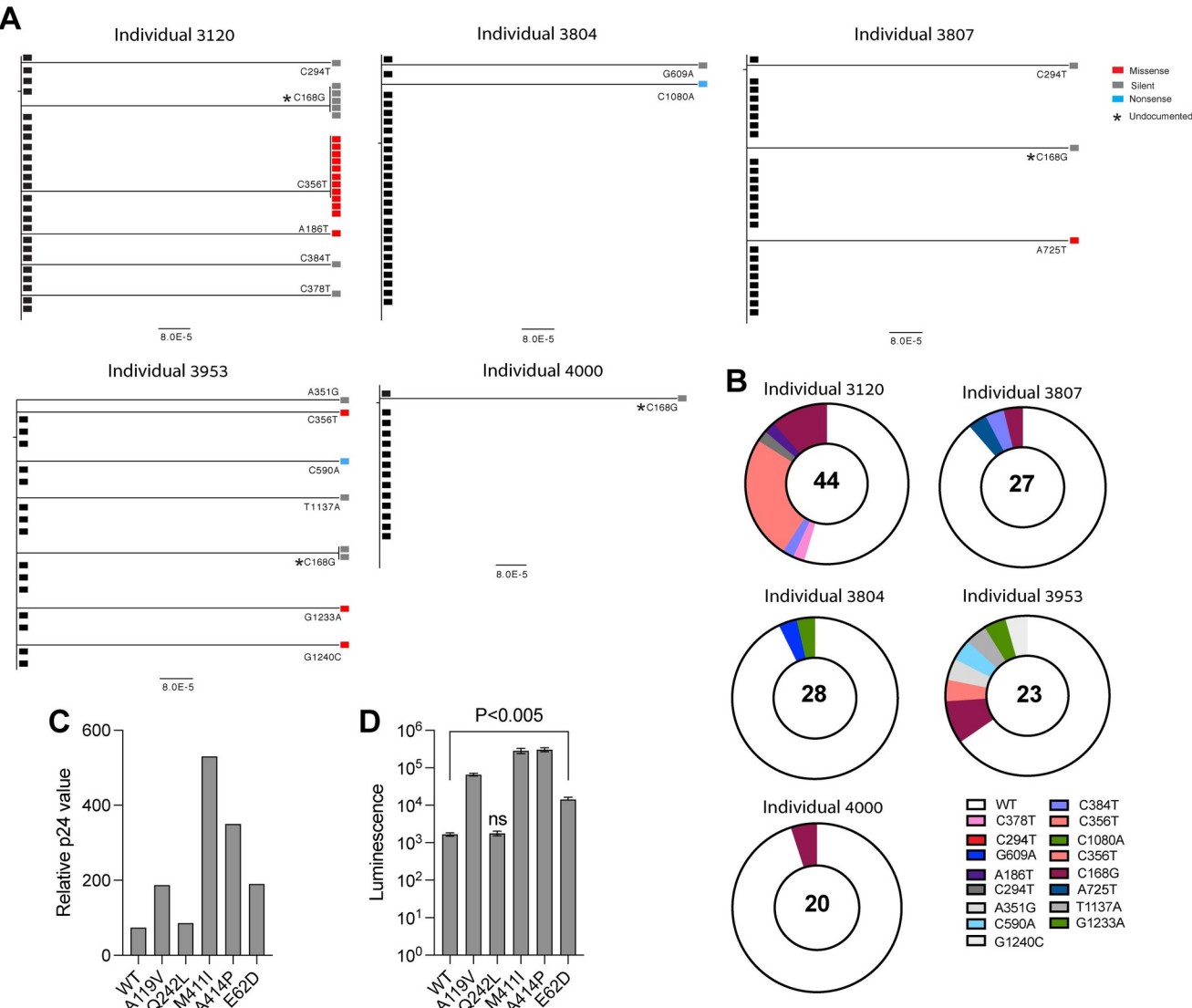

**Fig 3. Intra-host SARS-CoV-2 N-gene diversity.** (A) Phylogenetic trees depicting N genes isolated from nose-throat swabs by SGS. IDs of individuals from whom swabs were isolated are indicated in the figure. Black rectangles depict the dominant N gene isolated from each swab sample. Red rectangles depict N genes with missense mutations, grey rectangles depict N genes with silent mutations and blue rectangles depict N genes with nonsense mutations. Mutations that are not documented in the GSAID are marked with an asterisk. For each N-gene variant, the nucleotide substitution is indicated (based on Wuhan-Hu-1 numbering). The segment with the number below each phylogenetic tree shows the length of branch that represents an amount genetic change. The amount of genetic change is the number of nucleotide substitutions divided by the length of the N-gene sequence. (B) Pie charts showing the proportion of all N-gene sequences isolated from each swab sample. The White pie slice depicts unmutated sequences. The Colored pie slices depict mutated N genes. The different mutations are indicated in the figure. The number in the middle of the pie chart depicts the total number of N-gene variants that were sequenced. (C) Relative p24 value as measured by adding 20 μl of supernatant containing the pseudovirus to Lenti-X GoStix Plus. The x-axis depicts the SARS-CoV-2 pseudoviruses that were produced, and the y-axis depicts the relative p24 protein (GoStix values) in the supernatant of each pseudovirus. (D) 293T-ACE2 infection with SARS-CoV-2 pseudovirses expressing mutated N proteins and unmutated (Wuhan-Hu-1) N protein (WT). The x-axis depicts the mutation in the N proteins of the pseudovirus that was used for infection. The y-axis depicts the luminescence levels that were measured 48 hours post infection. Experiments were done in triplicates and repeated three times. One representative experiment is shown. Mean values and standard errors are shown. Statistically significant differences in comparison to the WT SARS-CoV-2 pseudovirus are indicated (student's t test, p < 0.005). Figure was generated using biorender.com.

showed significantly reduced infectivity compared to SARS-CoV-2 pseudovirus carrying unmutated (Wuhan-Hu-1) spike protein (Fig 4D). Of note, pseudoviruses that carry the mutation F543S at the S1 subunit and C840F mutation at the S2 subunit resulted in the most

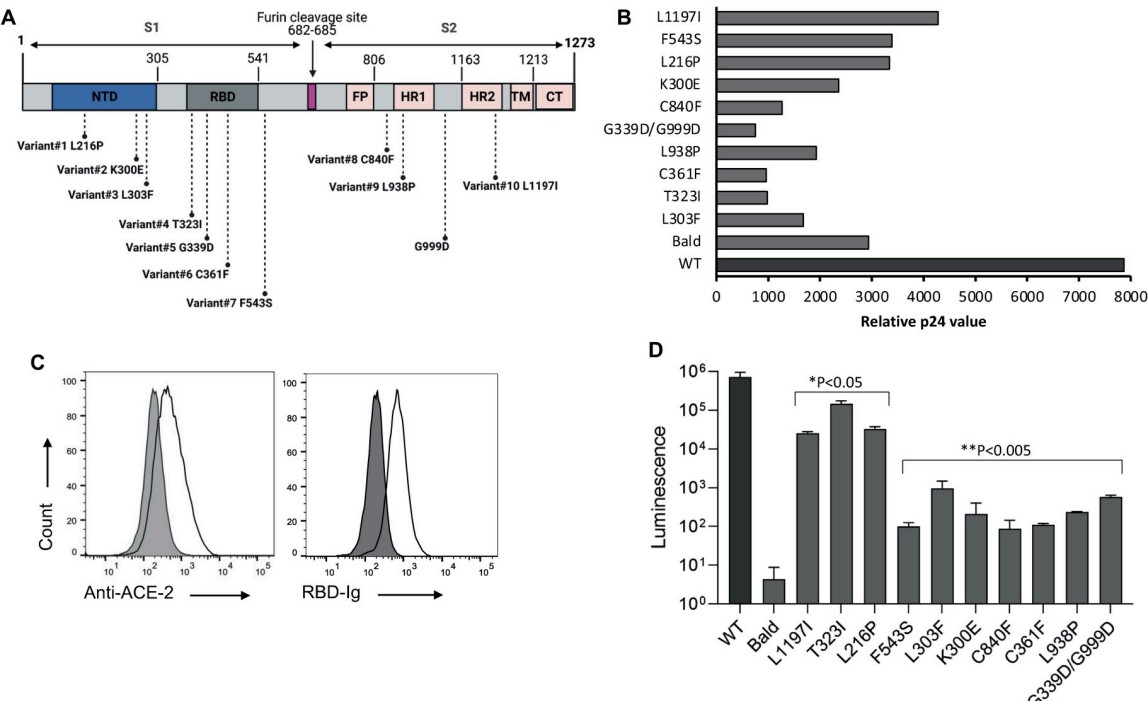

**Fig 4. Reduced infectivity of SARS-CoV-2 pseudoviruses expressing mutated spikes.** (A) Schematic representation of the ten spike variants that were tested for infectivity. Amino acids substitution in each spike mutants is shown in the figure. NTD = N-terminal domain, RBD = receptor binding domain, FP = fusion peptide, HR1 = heptad repeat 1, HR2 = heptad repeat 2, TM = transmembrane domain, CT = cytoplasmic domain. (B) Relative p24 value as measured by adding 20 µl of supernatant containing the pseudovirus to Lenti-X GoStix Plus. The y-axis depicts the SARS-CoV-2 pseudoviruses that were produced, and the x-axis depicts the relative p24 protein (GoStix values) in the supernatant of each pseudovirus. WT = pseudovirus that expresses an unmutated (Wuhan-Hu-1) spike. (C) FACS staining of 293T-ACE2 cells. The gray histogram shows the staining of the 293T-ACE2 cells with secondary antibody only. The empty black histograms depict the staining anti-ACE2 antibody or RBD-Ig. Shown is one representative experiment out of three preformed. (D) 293T-ACE2 infection with SARS-CoV-2 pseudovirses expressing mutated spikes, unmutated (Wuhan-Hu-1) spike protein and bald pseudovirus. The x-axis depicts the mutation in the spike of the pseudovirus that was used for infection. The y-axis depicts the luminescence levels that were measured 48 hours post infection. Experiments were done in triplicates and repeated three times. One representative experiment is shown. Mean values and standard errors are shown. WT = pseudovirus that expresses an unmutated (Wuhan-Hu-1) spike. Statistically significant differences in comparison to the WT SARS-CoV-2 pseudovirus are indicated (student's t test, $^*p < 0.05$, $^{**}p < 0.005$). Figure was generated using biorender.com.

significant reduction in infection, while the RBD mutation that also removed an N-glycosylation site (T323I) resulted in the most modest reduction in infectivity in comparison to all other spike mutations. We conclude that intra-host spike variants that are generated during the course of infection exhibit reduced infectivity in comparison with viruses that carry the Wuhan-Hu-1 spike protein.

## Intra-host spike variants demonstrate varied sensitivity to the BNT162b2 vaccine and to convalescent plasma

Previous studies demonstrated that viral mutations that promote resistance to neutralizing antibodies often impair the affinity of the viral protein to the cellular receptor and result in reduced infectivity of the mutated virus [23–26]. Thus, we next evaluated the interaction of the variants that showed reduced infectivity with neutralizing antibodies that are elicited by the BNT162b2 vaccine or following SARS-CoV-2 infection. To test this, we recruited 7 volunteers who had received 2 doses of the Pfizer–BioNTech mRNA vaccine BNT162b2 for blood donation two weeks after the second dose administration (S3 Table). First, we measured the level of

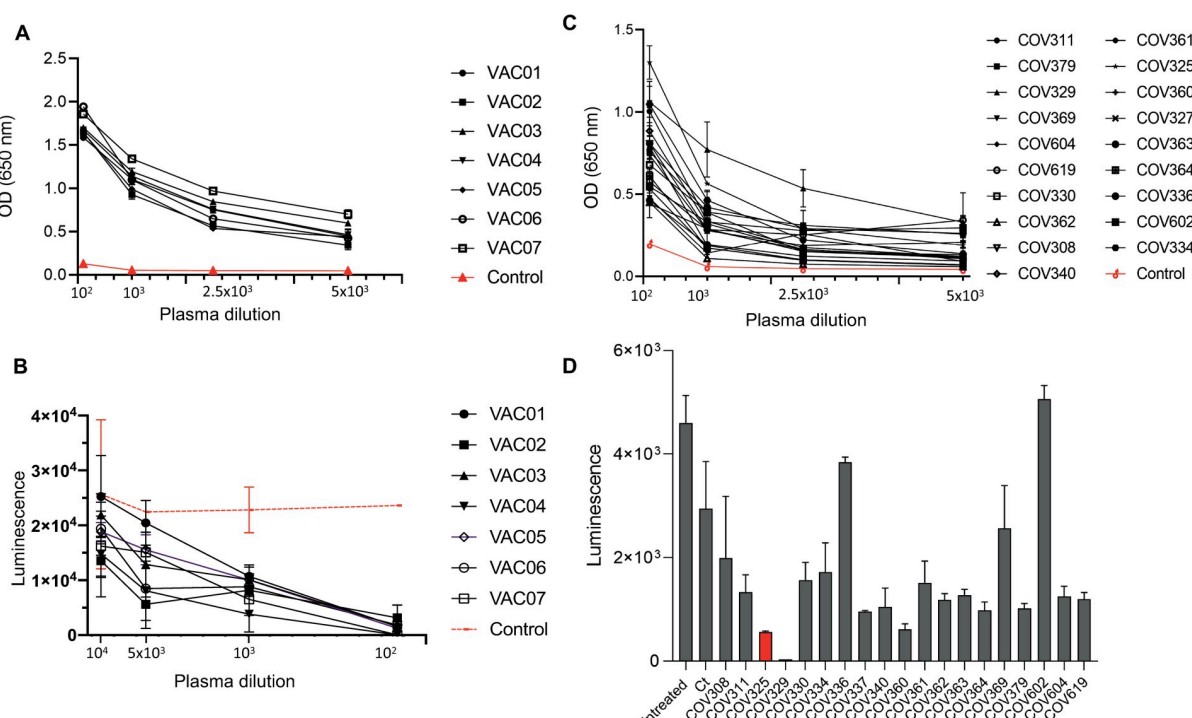

**Fig 5. Levels of anti-spike IgG and neutralization activity of plasma samples.** (A, C) OD values of ELISA against spike S1 subunit of plasma from vaccinated individuals (A) and of convalescent plasma (C). Experiments were done in triplicates and repeated three times. One representative experiment is shown. Mean values and standard errors are shown. OD values of control plasma (unvaccinated healthy individual) are shown in the red dashed graph. (B) The luminescence values derived from 293T-ACE2 cells 48 hours after infection with nanoluc-expressing SARS-CoV-2 pseudovirus and following incubation with increasing concentrations of plasma from vaccinated individuals. The plasma dilution is shown in the x-axis. Luminescence values after incubation with control (unvaccinated healthy individual) is shown in the red-dashed graph. Experiments were done in triplicates and repeated two times. Mean values and standard errors are shown; representative of two independent experiments is shown. (D) The luminescence values derived from 293T-ACE2 cells 48 hours post infection with nanoluc-expressing SARS-CoV-2 pseudovirus and following incubation with convalescent plasma at $10^{-3}$ dilution. The study ID of the plasma samples is shown in the x-axis. Experiments were done in triplicates and repeated two times. Mean values and standard errors are shown; representative of two independent experiments is shown.

anti-spike IgG antibodies in the plasma of the vaccinated individuals. All of the vaccinated individuals (n = 7) showed high levels of anti-spike IgG antibodies while no reactivity against the spike protein was seen in plasma from an unvaccinated individual (Fig 5A). Moreover, consistent with the presence of anti-spike IgG antibodies, all plasma samples showed neutralization activity against SARS-CoV-2 pseudoviruses that carry Wuhan-Hu-1 spike protein (Fig 5B). We also collected convalescent plasma samples from new volunteers that were not included in the SGS analysis (n = 20, S4 Table), measured the anti-spike IgG levels in those samples and evaluated their neutralization activity. All convalescent plasma samples were positive for anti-spike IgG antibodies (Fig 5C) and showed high variability in their neuralization activity against SARS-CoV-2 pseudoviruses (Fig 5D), probably due to the fact that these plasma samples were collected at different time points after SARS-CoV-2 exposure. Plasma samples COV325 and COV329 showed the highest neutralization activity against SARS-CoV-2 pseudoviruses (Fig 5D). This is in accordance with the fact that these individuals showed the highest levels of anti-spike IgG antibodies (Fig 5C).

The neutralizing activity of these plasma samples against three of the spike variants that we have isolated from the infected individuals was tested and showed the highest infectivity capacity: L216P, T323I and L1197I. When pseudoviruses that express the mutated spike proteins L216P and T323I were incubated with 10-fold dilutions of convalescent plasma or the

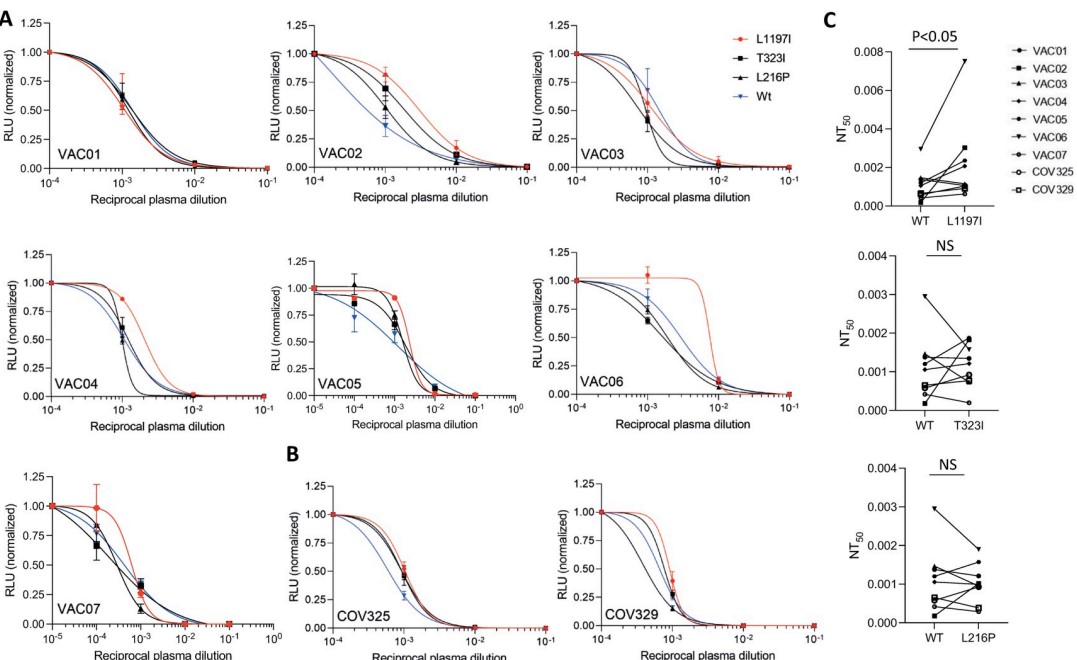

**Fig 6. SARS-CoV-2 pseudovirus neutralization assay.** (A-B) Neutralization assays, comparing the sensitivity of pseudotyped viruses with unmutated Wuhan-Hu-1 spike (WT) and the indicated spike mutations to plasma from vaccinated individuals (A) and convalescent plasma (B). Plasma dilutions are shown in the x-axis. The y-axis depicts the normalized relative luminescence units (RLU). Values were normalized to the RLU values seen with the $10^{-4}$ plasma dilution or the $10^{-5}$ dilution for VAC05 or VAC07. Mean values and standard errors are shown; representative of three independent experiments is shown. (C) $NT_{50}$ values for neutralization by plasma from vaccinated individuals (n = 7) and convalescent plasma (n = 2) against pseudotyped viruses with unmutated Wuhan-Hu-1 spike (WT) and the indicated spike mutations. Mean values and standard errors are shown. Statistically significant differences $NT_{50}$ values are indicated (student's t test, *p < 0.05). NS = not significant.

vaccinees' plasma, the neutralization activity that was observed was comparable to the neutralization activity against SARS-CoV-2 pseudoviruses that carry the Wuhan-Hu-1 spike protein (Fig 6A and 6B), and no significant different in the half-maximal neutralizing titer values ($NT_{50}$) was observed (Fig 6C). Notably, reduced neutralization activity of five out of the seven plasma samples from vaccinated individuals was seen against SARS-CoV-2 pseudoviruses that express the L1197I mutation at the HR2 domain of the spike S2 subunit (Fig 6A). Similarly, reduction in the neutralization activity against the L1197I variant was also seen in the two convalescent plasma samples that were tested (Fig 6B). Finally, when the $NT_{50}$ values of all plasma samples were compared, a significant increase was seen in the $NT_{50}$ values of plasma samples that were incubated with SARS-CoV-2 pseudoviruses carrying the L1197I mutation in comparison with plasma samples that were incubated with SARS-CoV-2 pseudoviruses that express an unmutated spike protein (Fig 6C). To further test these differences, we also repeated the neutralization assay using a newly generated SARS-CoV-2 pseudoviruses carrying the Wuhan-Hu-1 spike protein or the L1197I mutation and evaluated the $NT_{50}$ values using 3-fold plasma dilutions (S7 Fig). Similarly, to the results presented in Fig 6C, the L1197I mutation led to a significant increase in the $NT_{50}$ values (S7B Fig).

## Discussion

Since the first report of SARS-CoV-2 infection in December 2019, the virus has continued to evolve as it circulated in the human population [7]. Examination of emerging mutated SARS-CoV-2 variants facilitates the understanding of viral adaptation to the human host and the

selective pressure exerted by the human immune responses. Generation of full-length SARS-CoV-2 genome sequences at an unprecedented speed, together with genome database initiatives, has facilitated real-time analyses of SARS-CoV-2 genomes across different geographic regions and has enabled to monitor the frequency of mutated variants and their global spread [34,35]. Evaluation of genomes from swab-derived clinical isolates was quickly followed by experimental approaches where SARS-CoV-2 ability to escape therapeutic agents and the current vaccines is tested directly [9,36,37]. Most of these studies focus on the SARS-CoV-2 spike protein, since it is the main target of the currently developed vaccines and novel neutralizing antibodies that were isolated from COVID-19-convalescent individuals [38–40].

Thus far, mRNA-based vaccines were highly effective in preventing SARS-CoV-2 infection and cases of SARS-CoV-2 infection in vaccinated individuals are relatively scarce [41]. This has enabled to effectively control the pandemic in regions with high vaccination rates [22]. However, an emergence of a vaccine-resistant SARS-CoV-2 variant could potently impede future efforts to control the COVID-19 pandemic. Several studies have demonstrated the potential of SARS-CoV-2 to escape potent neutralizing antibodies [9,20,42]. By using a recombinant chimeric VSV/SARS-CoV-2 reporter virus, it was shown that specific mutations at the spike protein can significantly impair antibody binding [37,42]. Despite these alarming findings, all the mutations that conferred resistance to tested neutralizing antibodies were rarely found in naturally circulating SARS-CoV-2 variants [42], which indicates that these mutations might lead to a viral fitness deficit that impairs the viral spread in the human host [43]. In accordance with this, viral escape from antibody neutralization by other RNA viruses such as HIV-1 has often led to impaired infectivity of the mutated virus [23–26]. For example, Manrique et al. demonstrated that resistance to the broadly neutralizing antibodies 2F5 and 4E10 is difficult to achieve, since these antibodies select variants with impaired infectivity [26]. Similar results were later observed for other HIV-1 broadly neutralizing antibodies such as VRC01 and when CD4 nanoparticles were used for HIV-1 suppression [23,25]. Whether a variant with impaired infectivity could spread in the human population is still not clear. However, the employment of SARS-CoV-2 vaccines changes the landscape of circulating SARS-CoV-2 variants [22] and as variants with reduced sensitivity to neutralizing antibodies are expected to be positively selected, variants with reduced infectivity might become more prevalent. To identify such variants, we adapted a method that was originally developed by Salazar-Gonzalez et al. and Plamer et al. to study HIV-1 diversification [13,14,44] and used it for single-genome amplification and sequencing of swab-derived SARS-CoV-2 genes. The level of intra-host variability that we have identified in nine SARS-CoV-2 infected individuals was significantly lower than the level of complexity and diversification seen in more diverse viruses, such as in HIV-1-infected individuals. This low variability was expected since SARS-CoV-2 replication is much less error prone in comparison with the high error rate of the HIV-1 reverse transcriptase [44–47]. Additionally, HIV-1 infection leads to a chronic disease in which viral adaptation to the host occurs for several decades and viral mutations accumulate over time [48,49].

The low frequency of the intra-host variants identified in the swab samples has led us to test the effect of the spike mutations that these viruses express on SARS-CoV-2 infectivity. In accordance with the low frequency of these spike mutants in the nose-throat swabs, we demonstrated that SARS-CoV-2 pseudoviruses that express these mutated spike proteins show reduced infectivity. Strikingly, an opposite outcome was seen with the N-protein variants that we have isolated, which showed increased infectivity. Interestingly, the spike T323I mutation that also removes an N-glycosylation site, led to a modest reduction in infectivity compared with the other spike mutations that were tested, despite being located in the RBD. However, the effect of impaired N-glycosylation sites on the affinity of the spike protein to ACE2 should be tested further.

We chose to preform in-depth analysis on the spike mutations that led to the most modest reduction in infectivity, L216P, T323I and L1197I. SARS-CoV-2 pseudoviruses that carry the L216P or the T323I mutation showed similar sensitivity to plasma samples from vaccinated individuals and to convalescent plasma in comparison with SARS-CoV-2 pseudovirus that expresses an unmutated spike protein. However, when SARS-CoV-2 pseudovirus with the L1197I mutation was incubated with these plasma samples, increased $NT_{50}$ values were observed. We did not test the mechanism by which this HR2 domain mutation impairs antibody neutralization. However, this effect does not seem to be specific to a certain neutralizing antibody, since impaired plasma activity against these variants was seen in most of the plasma samples we have tested. As in-depth profiling of amino acid substations in the spike RBD that alter SARS-CoV-2 antibody neutralization was recently performed [50], our data suggests that amino acid substitutions at the spike S2 subunit and their effect on antibody neutralization should be further characterized.

The intra-host variants we have isolated from nose/throat swabs were all found in low frequency and were found to co-exist in the respiratory system together with a more dominant and highly frequent variant. Accordingly, when we tested the infectivity of ten of these variants, we found that the spike mutation that they carry impairs their infectivity. The exact mechanism that led to this reduced infectivity was not evaluated here. Other stages of the viral production should be tested such as the impact of the spike mutations on viral packaging and protein synthesis. Whether these variants can spread from person to person is not clear. However, we believe that mapping those mutations that impair SARS-CoV-2 infection can be used, together with structural analysis of the spike protein, to expose new vulnerability sites of this virus that will be used for achieving sustained suppression of SARS-CoV-2. Altogether, we conclude that genetic changes in SARS-CoV-2 that occur during the course of infection and in response to the host's selective pressure, need to be further explored in order to fully understand the COVID-19 pathogenesis.

The results of this study should be further evaluated using a larger cohort of individuals. This study included individuals who were infected with the D614G variants and two individuals who were infected with the Alpha variant (individual 48552 and 48559). It would be intriguing to analyze the intra-host variability of the Delta variant that is currently the dominant SARS-CoV-2 variant worldwide and to compare its intra-host diversity to the variants identified here. Moreover, the effect of additional spike mutations that recently emerged in the Delta variant such as the K417N [51] on SRAS-CoV-2 evolution within the host should be examined. Finally, by comparing our SGS results to bulk sequencing we revealed one of the limitations of the SGS results as several variants that were identify in the bulk sequencing were not identified by SGS. This is probably due to the limited number of viruses that were sequenced by our SGS approach. Thus, a higher sampling size is required to correctly estimate the variant landscape by SGS.

## Materials and methods

### Ethics statement

The study was approved by the Sheba Medical Center committee and Rambam medical center Institutional Review Boards and all swab and blood samples were obtained with written informed consent from the participants (Helsinki approval 7045-20-SMC, Helsinki approval 0060-21-RMB).

### COVID-19 swab samples and plasma samples

SARS-CoV-2 swab samples were derived from patients admitted to Sheba Medical Center, Israel (Helsinki approval 7045-20-SMC). SARS-CoV-2 infection was confirmed by reverse-

transcription quantitative polymerase chain reaction (RT-PCR) on nasopharyngeal swab samples. Swab samples were chosen based on the age of the infected individual and on the estimated infection region to minimize oversampling biases. Demographic and clinical data for patients are provided in Fig 1A. Viral loads were evaluated using RT-PCR reactions, using primers corresponding to the SARS-CoV-2 envelope (E) gene as previously described [27]. The qRT-PCR reactions were performed in 25 μl Ambion Ag-Path Master Mix (Life Technologies, USA) using TaqMan Chemistry on the ABI 7500 instrument. Blood samples from vaccinated individuals were taken two weeks after the second dose of the BNT162b2 vaccine at the Rambam Health Care Campus (Helsinki approval 0060-21-RMB). Peripheral blood was centrifuged at 500g for 8 minutes; heparinized plasma and serum samples were aliquoted and stored at -20˚C. Before the experiments, aliquots of plasma samples were heat-inactivated at 56˚C for 1 hour and then stored at 4˚C or lower. Convalescent plasma samples were purchased from Almog diagnostic's. To measure the viral load in the swab samples we adapted a method used to detect SARS-CoV-2 in clinical samples [52]. Viral load was calculated from the Ct values obtained by qRT-PCR, using a standard curve that was generated using 10-fold dilutions of standard RNA starting from 200,000 copies to 2,000 copies. Reaction was done in triplicates. The resulting Ct values for each sample were then plotted against the known RNA copies used in each dilution. Based on this curve, the viral loads of the samples used in this study were determined. Primers that target the N-gene were used for the qRT-PCR [52]: 5' primer 'CACATTGGCACCCGCAATC' and 3' primer 'GAGGAACGAGAAGAGGCTTG' [52].

## Cells and bacterial strains

HEK 293T cells stably expressing human ACE2 (293T-ACE2) were established in our lab by overexpression of human ACE2 using a lentiviral vector and repeated sorting of the ACE2-expressing cells. HEK-293T cells (ATCC, CRL-3216) and 293T-ACE2 cells were cultured in Dulbecco's Modified Eagle Medium (DMEM) supplemented with 4500 mg/l D-glucose, 4 mM L-glutamine, 110 mg/l sodium pyruvate, 10% FBS, 1% penicillin–streptomycin, and 1% nonessential amino acids (NEAA). Cells were maintained at 37˚C with 5% $CO_2$. DH5α bacteria (Thermo Fisher Scientific,18258012) grown in LB media at 37˚C were used for cloning and amplification of plasmid DNA for mammalian cell transfection.

## Single genome sequencing of SARS-CoV-2 genes

Viral RNA was isolated from SARS-CoV-2 swabs using MagLead 12gC (Precision System Science Co., Ltd., Japan), according to the manufacturer instructions. SuperScript III Reverse Transcriptase (Invitrogen, Cat #18080–044) and N-Out-3' primer were used for cDNA synthesis. cDNA was synthesized at 50˚C for one hour followed by an additional hour at 55˚C. The reaction was terminated by heat inactivation at 70˚C for 15 minutes. Remaining RNA was digested with RNaseH (Invitrogen, Cat #EN0201) for 20 minutes at 37˚C. cDNA was serially diluted and was subjected to two rounds of nested PCR of SARS-CoV-2 S gene and SARS-CoV-2 N gene of 12 cDNA dilutions using specific primers and PCR conditions (S1 Table). cDNA dilutions that provided lower than 30% amplification efficiencies were used for generation of S-gene and N-gene libraries. According to a Poisson distribution, the cDNA dilution that yields PCR products in no more than 30% of wells contains one amplifiable cDNA template per positive PCR more than 80% of the time [53]. The final PCR products were analyzed by 1% agarose gel electrophoresis, and amplicons of the correct size were used to prepare DNA libraries by the Illumina Nextera DNA Sample Preparation Kit as previously described [15,16]. All PCR reactions were performed using Platinum Taq HiFI (Invitrogen, Cat #11304–029) to ensure high fidelity PCR reactions. Sequencing was performed using the Illumina

Miseq Nano 300 cycle kits at a concentration of 12 pM. Gene alignments were generated using Geneious 9.1.8 (Biomatters). NC_045512 SARS-CoV-2 isolate Wuhan-Hu-1, complete genome was used as reference genome.

## Bulk sequencing of SARS-CoV-2 genes

cDNA from individuals 3120, 4000, 4065,3380 and 3807 was used as a template for nested PCR of the SARS-CoV-2 gene. The final PCR products were analyzed by 1% agarose gel electrophoresis, and amplicons of the correct size were used to prepare DNA libraries by the Illumina Nextera DNA Sample Preparation Kit as previously described [15,16]. All PCR reactions were performed using Platinum Taq HiFI (Invitrogen, Cat #11304–029) to ensure high fidelity PCR reactions. Sequencing was performed using the Illumina Miseq Nano 300 cycle kits at a concentration of 12 pM. Gene alignments were generated using Geneious 9.1.8 (Biomatters). NC_045512 SARS-CoV-2 isolate Wuhan-Hu-1, complete genome was used as reference genome. Variant calling analysis was performed to identify single nucleotide polymorphism (SNP) for each sample. The variant calling is conducted by the "HaplotypeCaller" tool of GATK [54]. This tool performs realignment around active regions to compose haplotypes. For each active region, the tool builds a De Bruijn-like graph to reassemble the region, and it identifies what the possible haplotypes present in the data are. The tool then realigns each haplotype against the reference haplotype using the Smith-Waterman algorithm to identify potentially variant sites. A matrix of likelihoods of haplotypes is produced. These likelihoods are then marginalized to obtain the likelihoods of alleles for each potentially variant site. Then the tool calculates the likelihoods of each genotype per sample, given the read data observed for that sample. The most likely genotype is then assigned to the sample.

## Flow cytometry analysis of ACE2 protein expression

293T-ACE2 ($5x10^4$) cells were seeded in a 96-well plate, washed, and were incubated with polyclonal anti-ACE2 antibody (Bioss, BS-23028R) or RBD-Ig [55] for 1 hour at 4˚C at 1:20 dilution, followed by incubation with PE-conjugated goat anti-human IgG (Jackson, Cat #109-116-088, 1:200) for 45 minutes. Control cells were stained only with PE-conjugated goat anti-human IgG. Cells were washed, and staining was evaluated using the BD LSRFortessa flow cytometer. Further analysis of the staining results was done using FlowJo software v10.7.

## Production of SARS-CoV-2 pseudoviruses and 293T-ACE2 infection

SARS-CoV-2 pseudoviruses carrying unmutated (Wuhan-Hu-1) spike protein or mutated spike protein were produced by co-transfection of 293T cells with 2.5 μg of pMD2.G SARS-CoV-2 –SΔ19-opt mixed with 7 μg of pPAX2 and 7 μg pLenti-Luc. For generating viruses that also contain the N-protein, the cells were transfected with 2.5 μg of pMD2.G SARS-CoV-2 –SΔ19-opt mixed with 7 μg of pPAX2, 7 μg pLenti-Luc and 3.75 pLenti that encodes the N-gene. 48 hours post transfection, the supernatants were harvested and centrifuged at $350 \times g$ for 5 minutes to remove cell debris and filtered through 0.22 μM filter. Supernatants were collected and were kept in -80˚C until use. Bald lentiviral pseudoviruses were generated by omitting pMD2.G SARS-CoV-2 –SΔ19-opt from the transfection mix. The presence of viral particles in the supernatants was verified by measuring p24 levels using Lenti-X GoStix Plus (NC1465071) according to the manufacturer's instructions. SARS-CoV-2 pseudoviruses and bald pseudoviruses were used to infect $2x10^4$ 293T cells or 293T-ACE2 cells that were seeded in triplicates in a 96-well plate. Polybrene (10μg/ml) was added to increase infectivity [36]. Infection was evaluated after 48 hours by using a one-step luciferase system (cat #60690–1 BPS Bioscience) and by analyzing luciferase activity using a plate reader (Infinite M200 PRO).

## ELISA

To evaluate the presence of IgG anti-spike antibodies in plasma samples, ELISA was performed by coating high-binding 96-well plates (Thermo scientific, Ca t#44-2404-21) with 50 ul per well of a 1μg/ml of the S1 subunit in sodium bicarbonate solution overnight at 4˚C. Plates were washed 6 times with washing buffer (1xPBS with 0.05% Tween-20) and incubated with 200 μl/well blocking buffer (1xPBS with 2% skim milk) for 2 hours at room temperature. Immediately after blocking, plasma samples were assayed at a 1:100 starting dilution and 4 additional serial dilutions in blocking buffer. Plates were incubated again for 2 hours at room temperature. After washing 6 times with washing buffer, plates were incubated with anti-human IgG secondary antibody conjugated to horseradish peroxidase (HRP) diluted in blocking buffer at 1:10,000 dilution. Plates were developed by the addition of the HRP substrate, TMB (Life technologies, Cat #002023) for 8 minutes. After that, the absorbance was immediately measured at 650 nm with an ELISA microplate reader (Infinite M200 PRO). Data were analyzed using Prism V8.0. Plasma dilutions were done using a blocking buffer.

## Pseudotyped virus neutralization assay

Serially diluted plasma samples ($10^{-1}$–$10^{-4}$) from COVID-19-convalescent individuals, a healthy donor and vaccinated individuals were incubated with SARS-CoV-2 pseudovirus carrying unmutated (Wuhan-Hu-1) spike protein or mutated spike protein for 1 h at 37˚C. Dilutions of the plasma samples were done using DMEM Dulbecco's Modified Eagle Media with no supplements. Values were normalized to the RLU values seen with the $10^{-4}$ plasma dilution. Additional plasma dilution of $10^{-5}$ was done for samples VAC05 and VAC07 and the RLU values were normalized to the RLU values seen with the $10^{-5}$ plasma dilution. The mixture was subsequently incubated with $2x10^4$ 293T-ACE2 cells in a 96-well plate for 48 hours at 37˚C 5% $CO_2$, after which infectivity/neutralization was checked using One-step luciferase system (cat #60690–1 BPS Bioscience) and by analyzing luciferase activity using a plate reader (Infinite M200 PRO). The half-maximal inhibitory concentrations for plasma samples ($NT_{50}$) were determined using a four-parameter nonlinear regression (GraphPad Prism).

## Supporting information

**S1 Fig. SGA PCR products analysis on 1% agarose gel.** cDNA from individual 3120 was diluted into 1:2500 dilution (SGA) and subjected to nested PCR for SARS-CoV-2 S gene. PCR products were loaded on E-Gel-96 1% agarose. Positive products of ~3.8 kb are detected in less the 30% of all wells.
(TIF)

**S2 Fig. SARS-CoV-2 spike and nucleocapsid mutations on the genome sequence.** (A) All spike mutations in SGA-derived variant isolated in these study. The amino acid position of each spike mutation is indicated in the figure. The color regions depicts the NTD (N-terminal domain), RBD (receptor-binding domain), FP (fusion peptide), HR1 (heptad repeat 1), HR2 (heptad repeat 2), TM (transmembrane domain) and CTD (C-terminal domain). (B) All nucleocapsid mutations in SGA-derived variant isolated in these study. The amino acid position of each nucleocapsid mutation is indicated in the figure. The color regions depicts the NTD (N-terminal domain), RBD (RNA-binding domain), LINK (central linker), Dimerization domain and CTD (C-terminal domain). Silent mutations in the SGA-derived variant are not shown.
(TIF)

**S3 Fig. Phylogenetic tree depicting Wuhan-Hu-1 spike genes analyzed by SGS.** 55 control Wuhan-Hu-1 spike genes were amplified by nested PCR using S primers listed in S1 Table. Blue rectangle depicts the WT Wuhan-Hu-1 spike gene. Black rectangles depict the S genes that were amplified.
(TIF)

**S4 Fig. Spike and nucleocapsid mutations frequency in GISAID database.** The frequency of the (A) S-gene nucleotide substitutions and (B) N-gene isolated in this study by Single Genome Sequencing (SGS) in the GISAID database. Frequency calculation was made based on sequencing data archived in GISAID by November 18, 2021.
(TIF)

**S5 Fig. N variants mRNA expression.** RNA was purified from 293T cells 48 hours post transfection with the indicated N variant. Purified RNA was reversed transcribed into cDNA and subjected to PCR using proper primers (Forward 5'-ATGAGCGACAACGGACCACA-3' and Reverse 5'-CTGTTGCAACTGCTTAGAGAAG-3').
(TIF)

**S6 Fig. SARS-CoV-2 pseudovirus infection level.** 293T cells and 293T cells expressing ACE-2 were infected with SARS-CoV-2 pseudovirus, and the luminescence levels (y-axis) were measured 48 hours post infection. Statistically significant differences were calculated using student's t test (***$p < 0.0001$).
(TIF)

**S7 Fig. SARS-CoV-2 pseudovirus neutralization assay.** (A) Neutralization assay, comparing the sensitivity of pseudoviruses with unmutated Wuhan-Hu-1 spike (WT) and the L1197I spike mutation to plasma from vaccinated individuals. The plasma dilutions factors (3-fold) are shown in the x-axis. The y-axis depicts the normalized relative luminescence units (RLU). Values were normalized to the RLU values seen with the $10^{-5}$ dilution. Mean values and standard errors are shown; representative of three independent experiments is shown. (B) $NT_{50}$ values for neutralization by plasma from vaccinated individuals (n = 7) against pseudoviruses, WT and L1197I. Mean values and standard errors are shown. Statistically significant differences $NT_{50}$ values are indicated (student's t test, *$p < 0.05$).
(TIF)

**S1 Table. Primers used for nested PCR of SARS-CoV-2 S-gene and N-gene.**
(TIF)

**S2 Table. Summary of Variant calling analysis.**
(TIF)

**S3 Table. BNT162b2 vaccinated individuals' data.**
(TIF)

**S4 Table. COVID-19 convalescent plasma donors' data.**
(TIF)

## Acknowledgments

We thank all study participants who provided samples for our research. We thank the TGC–Technion Genome Center for their sequencing services and genome analysis, especially Tal Katz-Ezov, Gur Sevillya and Maor Hatoel. Blood samples from individuals vaccinated with BNT162b2 vaccine were obtained from the Israeli National Biobank for Research (MIDGAM) at the Rambam Health Care Campus. Some of the figures were generated using biorender.com.

## Author Contributions

**Investigation:** Dina Khateeb, Tslil Gabrieli, Bar Sofer, Adi Hattar, Sapir Cordela, Ilia Spivak.

**Resources:** Abigael Chaouat, Izabella Lejbkowicz, Ronit Almog, Michal Mandelboim.

**Supervision:** Yotam Bar-On.

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
