## [Decision Letter · Decision Letter 0]

27 Sep 2021

Dear Dr Technion,

Thank you very much for submitting your manuscript "SARS-CoV-2 variants with reduced infectivity and varied sensitivity to the BNT162b2 vaccine are developed during the course of infection" for consideration at PLOS Pathogens. As with all papers reviewed by the journal, your manuscript was reviewed by members of the editorial board and by several independent reviewers. In light of the reviews (below this email), we would like to invite the resubmission of a significantly-revised version that takes into account the reviewers' comments.

The reviews by the two referees come to opposite conclusions as to the recommendation to publish in PLOS PATHOGENS. I am willing to ask for a revision, but ALL of the comments by the reviewers have to be adequately addressed and the limitations of the study have to be carefully spelled out.

Acceptance of the paper depends on a thorough revision!

We cannot make any decision about publication until we have seen the revised manuscript and your response to the reviewers' comments. Your revised manuscript is also likely to be sent to reviewers for further evaluation.

Sincerely,

Peter Palese

Associate Editor

PLOS Pathogens

Volker Thiel

Section Editor

PLOS Pathogens

Kasturi Haldar

Editor-in-Chief

PLOS Pathogens

orcid.org/0000-0001-5065-158X

Michael Malim

Editor-in-Chief

PLOS Pathogens

orcid.org/0000-0002-7699-2064

The reviews by the two referees come to opposite conclusions as to the recommendation to publish in PLOS PATHOGENS. I am willing to ask for a revision, but ALL of the comments by the reviewers have to be adequately addressed and the limitations of the study have to be carefully spelled out.

Acceptance of the paper depends on a thorough revision!

Reviewer's Responses to Questions

**Part I - Summary**

Reviewer #1: Khateeb et al. used single genome sequencing (SGS) to characterize sequence variations in the SARS-CoV-2 spike (S) and nucleocapsid (N) genes from nasal swab samples. SARS-CoV-2 variants that successfully transmit from person to person have been the subject of intense investigation. The degree of variation that occurs within an infected individual is also of interest, however, because this may provide insights into virus evolution. The present study makes a good start towards addressing in vivo variation. However, the study has several limitations that reduce its significance. Among these, the authors do not conclusively demonstrate that the SGS approach yields an accurate measure of variant frequency. The number of individuals sampled is small and the conclusions that can be drawn from this study are quite limited. Finally, some technical concerns regarding the neutralization experiments were noted.

Reviewer #2: This manuscript aims to examine the intra-host variation of SARS-CoV-2 in the context of viral sensitivity or resistance to vaccine induced antibodies. By using single genome sequencing method, the authors uncover the sequences of low-frequency viral variants that exist in swabs of seven infected donors, and are less likely to be detected by regular RT-PCR reactions. Interestingly, based on S and N proteins sequencing many of the infected donors exhibited quasi-species in additional to the “main” viral strain. As might be expected, many of the intra-host variants exhibited stop-codon mutations, making them nonfunctional, however a small number of variants had no apparent S or N- related defects, and of these some exhibited different levels of resistance to vaccine-induced antibodies.

As opposed to numerous other studies describing SARS-CoV-2 inter-host variation, where bulk viruses are sequenced, the topic of viral diversity within the same person is less addressed. Such analysis is interesting from both basic-virological and translational-medical aspects, and therefore this work is definitely worth publishing. There are several typos and inconsistencies between the text and the figures that must be corrected before publication.

**Part II – Major Issues: Key Experiments Required for Acceptance**

Reviewer #1: 1. The samples prepared for SGS were amplified by nested PCR. Were control PCR reactions performed to determine at what frequency variation is introduced by the PCR? The authors indicate that they use a high-fidelity polymerase for the PCR, but it would be reassuring to directly test PCR mutation frequency.

2. From any individual swab, up to roughly 40 S or N genes were sequenced. Therefore, even variants that are detected only once may represent about 2.5% or more of the total population. It would be helpful for the authors to perform next generation sequencing on each of the swab samples and determine at what frequency the SGS variants are detected. This could establish how well the SGS approach quantifies variant frequency. It would also test whether variants are missed by the SGS approach.

3. Figure 3 describes the results of the N gene sequencing. While the data are clearly described, it is unclear what conclusions can be drawn from these data, beyond the fact that variants can be identified. It would also be helpful to clarify why N was chosen for these experiments versus another gene(s).

4. Figure 5 assesses antibody responses in the individuals from whom swabs were taken for sequencing. The fact that individuals who harbor variants comprising a minor proportion of their viral load develop robust anti-spike antibody responses is not surprising.

5. In Figure 5, does the strong naturalization by COV325 and COV329 reflect higher levels of anti-spike antibodies or is there some qualitative difference that makes these sera better? This could be clarified by performing the assay with the same amount of anti-spike antibody from each sample.

6. The data in Figure 6 lead the authors to conclude that the L1197I mutation within spike S2 significantly impacts neutralization efficiency. However, performing this assay again but using a finer dose response curve is needed. The authors report 50% neutralization titers (NT5). However, many of the curves have only one data point that lies between 100% and 0% neutralization. This makes determining an accurate NT50 problematic. 3-fold or 5-fold serial dilutions would give more precise data than the 10-fold serial dilutions that are used.

Reviewer #2: none

**Part III – Minor Issues: Editorial and Data Presentation Modifications**

Reviewer #1: 1. Figure 2. The units for the scale bar beneath the phylogenetic trees should be defined.

2. Figure 3. The significance of the different colors in the pie charts should explained.

Reviewer #2: • I could not find in the manuscript how the viral load/titer was determined. Can the authors comment why the viral load/titer is different from Ct values for some samples in Fig 1B (for example for patients 4000 and 4065)?

• The authors did not address glycosylation issue – are some of the mutations they found affect addition or removal of glycan residues?

• Figure 2 is very hard to read and does not always consistent with the text. For example, in the text the authors write: “Further analysis of SARS-CoV-2 sequences from individual 3120 revealed S-gene variants with nonsense mutations (C1687T) in which a glutamine amino acid was replaced with a premature stop codon.“ however, looking at Figure 2, I was not able to find C1687T within the variants appearing in individual 3120, or any donor (I could find C1678T in donor 3953 – is this a typo?). Also, the authors write “Although the abundancy of each intra-host mutant was sparse, we also found a few S-gene mutations that were repeated in several viral isolates (nucleotide substitution A3300G and T1389C, Fig 2A).” Perhaps it is worth noting that the sequences repeated are all within the same donor 3807 and not shared between different donors as might be implied. The same is true for the bottom of page 6, the authors write: “S-gene variant C1686T that was isolated by our SGS approach was also found in 2x10-3 of the GISAID database” however, here too, I was unable to find variant C1686T in the Figure. The authors should make sure all the data in the Figure is consistent with the text. I also suggest that in Figure 2 the authors mark with different colors or shapes the variants/substitutions that exhibit nonsense mutations, as well at the common variants that appeared in more than one donor, or more than once in the same donor. The GISAID info can also be incorporated into Fig 2A to indicate which variant were [previously reported and which ones are new.

• Why for the analysis of the N protein the authors analyzed only five donors and not all seven? Should be explained in the text

• Figure 3: I was wondering why the representation of S (Figure 2) and N different?

• Figure 3B – are the same colors (red) represent similar mutation (C168T) repeating in the different individuals? I suggest putting a legend for the different colors and their mutations in 3B. Mutations that repeat in more than one donor should be indicated clearly. Also, I suggest that the numbers of the substitutions to be indicated on top of the sequence alignment in 3A.

• Similar Supplementary figure as the one made for S protein (Supplementary figure 2) should be prepared for N protein to summarize all mutations.

• More inconsistencies in the text – page 7 – the authors write that highest degree of variability was observed for donor 3120 who is 73 years old. However, in Figure 1 the clinical info states that this donor is 74 years old..

• Donor 3120 exhibited high sequence diversity in both N and S genes, did the authors check whether the degree of S and N diversity correlate with each other? Do the diversities of S and N correlate with viral load?

• The sentence in Page 7 “These variants were selected based on the mutations location and based on their absence or presence in the GISAID database “ is not clear.

• In Figure 4B it looks like all the S variants have lower p25 values and many of the variants have a dramatic 8-10-fold reduction in p24 levels compared to WT. Therefore, the S mutations could affect different stages of the viral production, such as packaging and protein synthesis. How can the authors be sure that the reduction in luminescence in 4D is due to reduced infectivity and not due to other mechanisms? Perhaps they should rephrase and use “reduced viability” instead?

• The authors probably submitted the manuscript before the emergence and sequence domination of the Delta VOC. Though the Delta VOC is mentioned once in the Introduction, I believe it will make the paper more relevant if the authors consider updating both the Introduction and the Discussion sections accordingly so they mention more the Delta variant, and the resistance mutations it encompasses, while citing the appropriate references. Also, the more recent nomenclature of the VOCs should be added (i.e., B117 is named “Alpha”, B351 “Beta”, P1 “Gamma”, etc.).

PLOS authors have the option to publish the peer review history of their article (what does this mean?). If published, this will include your full peer review and any attached files.

Reviewer #1: No

Reviewer #2: No
---

## [Decision Letter · Decision Letter 1]

23 Dec 2021

Dear Dr Technion,

We are pleased to inform you that your manuscript 'SARS-CoV-2 variants with reduced infectivity and varied sensitivity to the BNT162b2 vaccine are developed during the course of infection' has been provisionally accepted for publication in PLOS Pathogens.

Best regards,

Peter Palese

Associate Editor

PLOS Pathogens

Volker Thiel

Section Editor

PLOS Pathogens

Kasturi Haldar

Editor-in-Chief

PLOS Pathogens

orcid.org/0000-0001-5065-158X

Michael Malim

Editor-in-Chief

PLOS Pathogens

orcid.org/0000-0002-7699-2064

Reviewer Comments (if any, and for reference):

Reviewer's Responses to Questions

**Part I - Summary**

Reviewer #2: The authors did a fine job addressing all the critique. Aside from the typos in Figure 2, which they still need to correct, I do not have further queries.

Reviewer #3: In the revised manuscript by Khateeb et al., the authors have assessed the intra-host SARS-CoV-2 variants isolated from nose/throat swabs by performing single-genome sequencing. They demonstrated the co-existence of low-frequency SARS-CoV-2 variants and dominantly frequent strain. They also found that these low-frequency variants carry spike mutations that impair their infectivity. In addition, the sensitivity of mutated variants to neutralization by convalescent plasma and plasma from vaccinated individuals varied. Specifically, the L1197I mutation at the S2 subunit of the spike affects the plasma neutralization activity. It is worth mentioning that most previous studies described SARS-CoV-2 inter-host variation, while viral diversity within the same person, the topic of this manuscript, is less studied. Altogether, I agree with the authors that further analysis of SARS-CoV-2 quasispecies could be important for understanding the evolution of this virus, which might be relevant for the understanding of ongoing SARS-CoV-2 pandemics.

**Part II – Major Issues: Key Experiments Required for Acceptance**

Reviewer #2: (No Response)

Reviewer #3: The manuscript was reviewed by two reviewers who raised several issues the resolution of which required new experiments. In my opinion, the authors have successfully dealt with most of the reviewers' criticisms and revised the manuscript accordingly.

**Part III – Minor Issues: Editorial and Data Presentation Modifications**

Reviewer #2: (No Response)

Reviewer #3: (No Response)

PLOS authors have the option to publish the peer review history of their article (what does this mean?). If published, this will include your full peer review and any attached files.

Reviewer #2: No

Reviewer #3: No

---

## [Editor Report · Acceptance letter]

10 Jan 2022

Dear Dr Bar-On,

We are delighted to inform you that your manuscript, "SARS-CoV-2 variants with reduced infectivity and varied sensitivity to the BNT162b2 vaccine are developed during the course of infection," has been formally accepted for publication in PLOS Pathogens.

Best regards,

Kasturi Haldar

Editor-in-Chief

PLOS Pathogens

orcid.org/0000-0001-5065-158X

Michael Malim

Editor-in-Chief

PLOS Pathogens

orcid.org/0000-0002-7699-2064